# PI3K–AKT-Targeting Breast Cancer Treatments: Natural Products and Synthetic Compounds

**DOI:** 10.3390/biom13010093

**Published:** 2023-01-02

**Authors:** Yeqin Yuan, Huizhi Long, Ziwei Zhou, Yuting Fu, Binyuan Jiang

**Affiliations:** 1Medical Research Center, The Affiliated Changsha Central Hospital, Hengyang Medical School, University of South China, Changsha 410004, China; 2School of Pharmacy, Hengyang Medical School, University of South China, Hengyang 421001, China; 3Department of Clinical Laboratory, The Affiliated Changsha Central Hospital, Hengyang Medical School, University of South China, Changsha 410004, China

**Keywords:** PI3K–AKT pathway, breast cancer, natural products, clinical trial drugs

## Abstract

Breast cancer is the most commonly diagnosed cancer in women. The high incidence of breast cancer, which is continuing to rise, makes treatment a significant challenge. The PI3K–AKT pathway and its downstream targets influence various cellular processes. In recent years, mounting evidence has shown that natural products and synthetic drugs targeting PI3K–AKT signaling have the potential to treat breast cancer. In this review, we discuss the role of the PI3K–AKT signaling pathway in the occurrence and development of breast cancer and highlight PI3K–AKT-targeting natural products and drugs in clinical trials for the treatment of breast cancer.

## 1. Introduction

Cancer is a leading cause of death worldwide. In 2020, breast cancer surpassed lung cancer as the most common cancer type, accounting for 11.7% (2,261,419 cases) of all newly diagnosed cancer cases and 6.9% (684,996 cases) of all new cancer deaths worldwide [1]. Modern lifestyle has been associated with problems such as decreased fertility and obesity. These factors have contributed to the rising global incidence of breast cancer. The emotional and physical toll that conventional breast cancer treatment takes on survivors can be indelible. Although there currently appears to be no difference in postoperative complications between breast-conserving therapy (BCT) and mastectomy, this may depend on the combination of breast-conserving therapy and postoperative adjuvant therapy [2]. Radiation therapy to the marginal junction after lumpectomy is one of the methods of postoperative adjuvant therapy [3]. However, adjuvant therapy for patients rejected by radiotherapy continues to attract more attention. The phosphoinositide 3-kinase (PI3K)-protein kinase B (AKT) pathway is active in most breast cancers, and 40% of ER+ breast cancers are associated with PIK3CA mutation, which is a common genomic alteration in breast cancer [4]. In addition, the PI3K–AKT pathway, as a receptor tyrosine kinase (RTK) downstream pathway, participates in various cell biological functions such as breast cancer cell proliferation and apoptosis [5]. Targeting the PI3K–AKT pathway in breast cancer may be a potential drug target in the context of endocrine and anti-RTK resistance [6]. However, chemotherapy resistance and drug toxicity have been a concern in neoadjuvant therapy. In recent years, numerous studies have shown that natural products possess anticancer properties. Natural products and their derivatives account for 80% of FDA-approved anticancer drugs [7]. The PI3K–AKT signaling pathway has been associated with various tumorigenic functions in breast cancer, including cell survival, migration, and invasion [8]. In this review, we searched PubMed, Web of Science, clinicalTrials.gov, and China Knowledge Infrastructure (CNKI) for literature and clinical trial studies, and summarized natural products based on the PI3K–AKT pathway in breast cancer treatment, involving flavonoids, polyphenols, alkaloids, and PI3K inhibitors and AKT inhibitors in clinical trials. Finally, we discuss future directions and limitations of natural products.

## 2. PI3K–AKT Signaling in Breast Cancer

### 2.1. Overview of the PI3K–AKT Signaling Pathway

The PI3K–AKT cell signaling pathway is associated with the occurrence of various diseases, including neurodegenerative diseases, cancer, and tuberous sclerosis complex. Studies have shown that the binding capacity of PI3K and platelet-derived growth factor (PDGF) receptors affect AKT activation [9,10,11]. It has been demonstrated that AKT is activated by cytokines in a PI3K-dependent manner [12,13,14]. Moreover, AKT was the first effector substrate of PI3K to be identified [15]. PI3K activation is related to the coordinated activation of multiple RTK like human epidermal growth factor receptor 2 (HER2), HER3, and insulin-like growth factor 1 receptor (IGF1R), but calmodulin is also involved [16]. RTKs are high-affinity cell surface receptors for various growth factors, cytokines, and hormones. These RTKs possess an extracellular ligand-binding domain, a transmembrane domain, and an intracellular tyrosine kinase domain [17]. The binding of ligands, such as growth factors, to the extracellular ligand domains of RTKs, triggers the two RTK monomers to dimerize, thereby activating the intracellular tyrosine kinase domain via the autophosphorylation of each monomer [18]. The regulatory subunit of PI3K binds to the intracellular tyrosine kinase domain and recruits the catalytic subunit to form a complex with the activated RTK [19]. The process plays a key role in the regulation of signaling through growth factors and cytokines. Additionally, the catalytic subunit p110 (α, δ, γ) directly binds to the rat sarcoma virus (RAS) family of GTPases to activate PI3K [20]. Activated PI3K catalyzes the phosphorylation of phosphatidylinositol 4,5-bisphosphate (PIP2), thereby generating the second messenger, phosphatidylinositol 3,4,5-trisphosphate (PIP3) [21]. The activity of class I PI3K is countered by the phosphatase PTEN (phosphatase and tensin homolog), which inhibits PI3K–AKT signaling by dephosphorylating PIP3 [22]. The ablation of PI3K’s catalytic subunit suppresses prostate carcinogenesis that results from PTEN deletion [23]. PIP3 accumulation on the plasma membrane recruits effector proteins such as AKTs. Differences in the functions of the three AKT subtypes stem from their differences in cellular distribution. AKT1 is widely distributed in tissues and plays an important role in cell growth and survival [24]. AKT2 is highly expressed in adipose and skeletal muscle and plays a specific role in the regulation of glucose homeostasis [25]. Compared with the previous two types, the distribution of AKT3 is smaller, mainly in the testis and brain. According to clinical studies, AKT3 mutations cause megalencephaly in many patients [26]. However, its specific pathogenic mechanism is still unclear. AKT1 is recruited to the plasma membrane through the interaction between its pleckstrin homology (PH) domain and PIP3, which leads to AKT1 phosphorylation at T308 by phosphoinositide-dependent protein kinase 1 (PDK1) [15]. This phosphorylation is essential for AKT1 activation and precedes subsequent AKT1 phosphorylation at S473 by mechanistic target of rapamycin (mTOR) complex 2 (mTORC2), to fully activate AKT [27,28,29].

The role of PI3K/AKT signaling in cells is positively correlated with the kinase activity of AKT. Glycogen synthase kinase-3 (GSK-3), the first AKT substrate to be identified, phosphorylates glycogen synthase in response to stimulation by insulin, thereby inactivating it [14]. mTOR is a key PI3K–AKT signaling substrate. AKT activates mTOR by phosphorylating it on S2448. Once activated, mTOR senses cellular energy and nutritional status, and then triggers responses such as autophagy, which generates sources of cellular energy [30]. AKT phosphorylates the protein tyrosine phosphatase (PTP1B), thereby negatively regulating its phosphatase activity and promoting insulin signaling [31]. AKT phosphorylates bcl2-antagonist of cell death (BAD), a member of the B-cell lymphoma 2 (Bcl-2) family, triggering its release from the mitochondrial membrane and preventing its binding to Bcl-2, thereby inhibiting its antiapoptotic function. Thus, by phosphorylating BAD, AKT promotes cell survival [32,33,34]. Studies have also shown that transcription factors, including the Forkhead (FH or FoxO) family of transcription factors and the nuclear transcription factor-kb (NF-κB)/Rel family are regulated by AKT [35]. AKT suppresses the activity of FoxO transcription factors via phosphorylation, thereby influencing the expression of its target genes, including TRADD (TNF receptor type 1-associated death domain) and intracellular apoptotic components such as Bcl-2 interacting mediator of cell death (Bim), and promoting proliferation [36]. NF-κB is a key regulator of immune responses. Tumor necrosis factor-α (TNF-α) activates AKT, which phosphorylates iκB kinase (IKK) (an upstream target of NF-κB) and activates NF-κB [37]. p21 is a cell cycle inhibitory protein (Cip1); the phosphorylation of p21 (T145) by AKT inhibits the nuclear localization of p21, attenuates the cyclin-dependent kinase 2 (Cdk2) inhibitory activity of p21, and promotes endothelial cell proliferation [38]. Figure 1 shows the role of the PI3K–AKT pathway in tumor cells.

### 2.2. The Role of PI3K–AKT Signaling in Breast Cancer

Important risk factors for breast cancer include genetic susceptibility, aging, hormone disorders, family history, and environmental factors. Others risk factors include physiological factors such as early menarche, late menopause, and obesity, as well as personal habits such as drinking and lack of physical activity [39,40]. The main types of breast cancer are lobular carcinoma in situ and ductal carcinoma in situ, with the latter being the most common. Breast cancer falls into five subtypes depending on the presence or absence of estrogen receptors (ER), progesterone receptors (PR), and HER2 receptors. These are the luminal A subtype (ER+, PR±, HER2−, low Ki67), luminal B subtype (ER+, PR±, HER2±, high Ki67), HER2 subtype (ER−, PR−, HER2+), triple negative (ER−, PR−, HER2−), normal-like (ER+, PR±, HER2−, low Ki67) [41,42,43].

The occurrence and development of breast cancer are driven by gene mutation and dysregulation of cell signaling pathways, and PI3K–AKT signaling is the most commonly upregulated pathway in breast cancer [44]. Several studies have implicated PI3K in the development of ER+ breast cancer. For example, the accumulation of PI3K gene mutations in somatic cells also promotes the development of breast cancer and PIK3CA mutations have been found in solid tumors [45]. Additionally, PIK3CA mutations suppress the differentiation of luminal and basal mammary cells, resulting in more cancer cell lineages [46]. John et al. found that mutation of E17K-Akt1 in human breast cancer cells can form a new hydrogen bond between Akt1 and phosphoinositol ligand, thereby activating Akt1 and promoting downstream signaling [47]. A meta-analysis found that breast cancer tissues have a higher rate of PTEN deletion when compared with normal tissues and that these mutations are associated with breast cancer invasiveness and metastatic potential. Patients with such mutations also have poor overall survival (OS) and disease-free survival (DFS) [48].

PI3K–AKT signaling activates ERα in an estrogen-independent manner and AKT overexpression protects breast cancer cells from tamoxifen (anti-estrogenic effect)-induced apoptosis [49]. This shows that inhibiting PI3K can enhance the therapeutic effect against ER+ breast cancer cells. Interestingly, it is reported that the p110α subunit of PI3K promotes apoptosis in estrogen-deprived breast cancer cells and that similar results are obtained using BEZ235, a PI3K inhibitor [50]. A PI3K gene expression signature enrichment analysis of ER+ breast cancer compared the two luminal breast cancer subtypes and found that the luminal B subtype has higher PI3K activity, which may make it less effective for antiestrogen therapy [42,51]. This also suggests that identifying the activation signature of the PI3K pathway can help predict the outcomes of endocrine therapy and identify high-risk ER+ breast cancers [52]. Additionally, it is reported that inhibiting mTOR, which acts downstream of AKT, enhances the antitumor effects of HER2 inhibitors in HER2-overexpressing breast cancer cells [53]. However, HER2/HER3 overexpression attenuates the antitumor effects of PI3K inhibitors [54]. Hence, to treat breast cancer, HER2 antagonists are often used in combination with PI3K inhibitors. DNA-damaging chemotherapy is often used to treat triple-negative breast cancer (TNBC). DNA damage often triggers the DNA-dependent protein kinase-mediated phosphorylation of AKT. Interestingly, several studies indicate that the pro-apoptotic effects of DNA-damaging agents are enhanced by PI3K inhibitors [55,56]. Therefore, PI3K pathway mutations may contribute to TNBC’s resistance to DNA-damaging agents. Glycogen synthase kinase-3β (GSK-3β) knockdown attenuated the apoptotic effect of rapamycin and paclitaxel on breast cancer cells [57]. This indicates that GSK-3β is a tumor suppressor in breast cancer. Taken together, these reports indicate that PI3K–AKT signaling has a key role in the development and treatment of various breast cancer subtypes.

## 3. Research Status of Inhibitors Targeting PI3K–AKT Pathway in Breast Cancer

Targeted therapy is an important strategy for cancer treatment. Table 1 is a summary of PI3K–AKT-targeting drugs that are in clinical trials (https://clinicaltrials.gov/ct2/home, accessed on 15 May 2022).

### 3.1. PI3K Inhibitors

PI3Ks fall into three classes based on primary structure and in vitro lipid specificity [81]. Class I PI3K is the most frequently studied class of enzymes that are activated by cell surface receptors and is further divided into class IA and class IB. Class IA PI3Ks are heterodimers of a p110 catalytic subunit and a p85 regulatory subunit. The p110 catalytic subunit is made of three homologous class IA catalytic isomers, p110α, p110β, and p110δ, encoded by PIK3CA, PIK3CB, and PIK3CD genes, respectively. The regulatory subunit of P85 is made of P85α (and its spliced variant P55α), P85β, and P55γ, which are encoded by PIK3R1, PIK3R2, and PIK3R3 genes [20,82]. Class IB PI3Ks are heterodimers of the catalytic subunit p110γ and the regulatory subunit p101 [83]. Class IA PI3Ks (α, β, δ) are activated upon the binding of the SH2 domain of the p85 regulatory subunit to activated RTK receptors or phosphotyrosine residue adaptor proteins, while class IB PI3Ks can be activated by G protein coupled receptors (GPCRs) [84]. Class II PI3Ks consist of a single subunit and are divided into three subtypes, PI3KC2α, PI3KC2β, and PI3KC2γ, which can be activated by various factors, including hormones, growth factors, and calcium ions. However, little is known about their functions [20]. Class III PI3Ks consist of a single subunit, vacuolar protein sorting 34 (VPS34), and are the only PI3Ks that are expressed in all eukaryotic cells. In vivo, Vps34 only phosphorylates phosphatidylinositol, generating phosphatidylinositol (3)-phosphate (PtdIns3P). VPS34 modulates mTOR signaling through its lipid kinase activity, thereby regulating cell growth [85]. VPS34 is also a key modulator of endosomal trafficking and autophagy.

A variety of PI3K inhibitors with the potential to treat breast cancer have been developed. Buparlisib (BKM120) is a potent, highly specific oral pan class I PI3K inhibitor. In clinical trials on the efficacy of BKM120 against advanced triple-negative breast cancer, Ana C et al. found that, alone, it prolonged SD in some patients, and that in combination with other drugs, it should be able to achieve better results [86]. Clinical trials on the efficacy of BKM120 against breast cancer, when combined with other drugs, such as lapatinib (NCT01589861), fulvestrant (NCT01339442), tamoxifen (NCT02404844), and LDE225 (NCT01576666), are ongoing. Interestingly, studies have shown that BKM120 sensitizes BRCA-proficient TNBC to poly ADP-ribose polymerase (PARP) inhibitors by downregulating the expression of BRCA1/2 and Rad51 via the inhibition of PI3K–AKT–NF-κB–c-Myc and the PI3K–AKT–forkhead box m1 (FOXM1)–Exonuclease 1 (Exo1) pathways [87].

Tenalisib is a highly selective, orally active dual PI3K δ/γ inhibitor that also possesses salt-inducible kinase 3 (SIK3) activity. A phase I/Ib study of tenalisib’s maximum tolerated dose (MTD), pharmacokinetics, and efficacy in patients with relapsed/refractory peripheral and cutaneous T-cell lymphoma found that it was clinically safe and tolerable at an MTD of 800 mg/day [88]. Additionally, ongoing studies on the use of tenalisib in combination with romidepsin to treat T-cell lymphoma indicate that tenalisib is well tolerated. [89]. Clinical trials on the use of tenalisib alone to treat patients with metastatic and invasive breast cancer are ongoing (NCT05021900).

Taselisib, in combination with fulvestrant, has previously undergone phase III clinical trials for the treatment of ER+, HER−, PIK3CA-mutated, advanced breast cancer. The results showed that although the taselisib group had a higher incidence of adverse events, INV-PFS (progression-free survival), objective response rate, clinical benefit rate, and duration of objective response after the combination showed a sustained improvement [90]. A phase III study on the use of taselisib in combination with fulvestrant to treat advanced breast cancer during aromatase inhibitor therapy or in relapsed disease is ongoing (NCT02457910).

BYL719 is a selective oral inhibitor of class I PI3K p110α. The combination of BYL719 and letrozole in the treatment of postmenopausal patients with ER+/HER− metastatic breast cancer that was refractory to endocrine therapy revealed that the combination was safe, tolerable, and effective, and phase II clinical trials showed that its MTD and recommended dose was 300 mg/day [91]. A study on the use of BYL719 in combination with paclitaxel to treat HER2− metastatic breast cancer found that patients with tumor/ctDNA mutations experienced better PFS when compared with patients without PIK3CA mutations, and that PFS was also higher in patients with normal metabolism [92]. Phase I clinical trials are ongoing on the use of BYL719 in combination with letrozole in the treatment of postmenopausal women with hormone receptor-positive metastatic breast cancer (NCT01791478).

Pictilisib (GDC-0941) is a potent and highly specific oral PI3K inhibitor (IC50: 3nM [93]. A study on the efficacy of pictilisib when combined with paclitaxel to treat hormone receptor positive, HER2−, locally recurrent, or metastatic breast cancer did not find significant PFS improvement in either the intention-to-treat or the PIK3CA-mutant groups [94]. In the subsequent phase Ib trials of pictilisib combined with paclitaxel, pictilisib + paclitaxel + bevacizumab, and pictilisib + paclitaxel + trastuzumab in the treatment of advanced breast cancer, the results showed that these three combinations had more controllable safety and anticancer activity [70,95].

GDC-0084, a dual PI3K/mTOR inhibitor, exhibited antitumor activity in preclinical models of glioblastoma [96]. In vitro treatment of PIK3CA-mutant breast cancer brain metastases cell lines with GDC-0084 reduced their viability, induced apoptosis, and inhibited the phosphorylation of AKT and p70S6, highlighting its potential as a therapeutic strategy against breast cancer with brain metastases [97]. A phase II clinical trial of the use of GDC-0084 in combination with trastuzumab to treat HER2+ breast cancer with brain metastases is underway (NCT03765983). Gedatolisib (PF-05212384) is a dual PI3K/mTOR inhibitor that also selectively inhibits PI3Kα and PI3Kγ with IC50 values of 0.4 nM and 5.4 nM, respectively [98]. Gedatolisib inhibits the growth of xenografted breast cancer at doses of >10 mg/kg [99]. Clinical trials are ongoing on the use of gedatolisib in combination with docetaxel, cisplatin, and dacomitinib, in triple negative breast cancer (NCT01920061).

AZD8186 is a potent PI3Kβ inhibitor that suppresses PI3Kδ signaling. AZD8186 combined with docetaxel showed a significant inhibitory effect on tumor in PTEN-deficient nude mice [100]. Moreover, AZD8186 is currently used in combination with docetaxel to treat patients with metastatic or unresectable PTEN- or PIK3CB-mutant advanced solid tumors, including breast cancers (NCT03218826).

Serabelisib is a potent, selective, oral PI3Kα inhibitor. A phase Ib study of the use of serabelisib (TAK117) in combination with sapanisertib (TAK228) and paclitaxel, to treat advanced ovarian, endometrial, or breast cancers, found that its dose and timing were well tolerated and that it was clinically active in paclitaxel-resistant patients [101]. Clinical trials on the use of serabelisib in combination with canagliflozin to treat breast cancer are ongoing (NCT04073680).

### 3.2. AKT Inhibitors

AKT has three highly homologous isoforms, Akt1 (PKBα), Akt2 (PKBβ), and Akt3 (PKBγ). AKT is comprises 480 amino acids and each isoform contains a conserved N-terminal PH domain (ATP binding), a kinase domain (catalytic function), and a C-terminal regulatory disordered tail (C-tail) [102]. Point mutations on the PH domain can limit AKT interaction with PIP3 and PIP2, thereby affecting its upstream kinase recognition and membrane translocation. AKT activation requires the phosphorylation of a threonine residue (AKT1-T308, AKT2-T309, and AKT3-T305) in the kinase domain and the phosphorylation of a serine in the C-terminal hydrophobic motif (AKT1-S473, AKT2-S474, and AKT3-S472). It has been shown that cyclin-dependent kinase 2 can activate AKT1 by phosphorylation of two C-terminal amino acids, S477 and T479 [103,104]. The three AKT isoforms regulate a wide range of functions, including cell growth, survival, proliferation, metabolism, and drug response.

MK-2206 is a selective allosteric inhibitor of AKT. Most PIK3CA-mutant or PTEN-deficient breast cancer cell lines are highly sensitive to MK-2206. A clinical trial found that MK-2206 inhibits AKT phosphorylation in platelets [105]. Results from this phase II clinical trial using MK-2206 alone found that it had limited clinical activity in patients with advanced breast cancer bearing PIK3CA or AKT mutations and PTEN deletion. [106]. When used in combination with anastrozole to treat PIK3CA-mutant, ER+/HER2− breast cancer, MK-2206 was unlikely to increase the efficacy of anastrozole [107]. In the phase I clinical trial involving patients with rectal cancer, ovarian cancer, and metastatic breast cancer, several patients completed dose escalation without experiencing dose-limiting toxicity. Moreover, post-treatment analysis revealed significantly reduced levels of phospho-AKT (S473 and T308), and that the combination was well tolerated [108]. Although the combination of MK-2206 and lapatinib in the treatment of solid tumors was well tolerated, the overlapping toxicity of the two drugs resulted in a high incidence of rash and diarrhea. However, this toxicity can be managed with drugs [109].

The above PI3K inhibitors and AKT inhibitors play an important role in the treatment of breast cancer by targeting the PI3K–AKT pathway. Compared with AKT inhibitors, the types of PI3K inhibitors undergoing clinical trials are more diverse. PI3K/mTOR dual inhibitors can target the kinase pockets of PI3K and mTOR due to their structural similarity [110]. However, mTOR inhibitors may enhance PI3K/PDK1 signaling, thus inhibitors targeting both PI3K and mTOR may have better anticancer activity [111]. Studies have shown that AKT inhibitors such as MK-2206, AZD5363, and GDC-0068 have increased activity in cell lines with altered *PIK3CA* or *PTEN* [112,113,114]. This suggests that AKT-specific inhibitors may exhibit greater potency in PTEN-altered tumors. As mentioned above, the occurrence of breast cancer is related to the cascading drive of the PI3K pathway and multiple signaling pathways. Combination RTK inhibitor therapy may be a good approach, for example with inhibitors of EGFR, HER2, or HER3 [115,116] or in combination with MEK inhibitors to overcome parallel induction of the MAPK pathway, a strategy that works well for PI3K pathway gene alterations and KRAS mutations. However, the improved efficacy of this strategy may increase the side effects of chemotherapy [117]. A preclinical study shows that CDK 4/6 inhibitors increase the sensitivity of PIK3CA-mutant breast cancer cells to PI3K inhibitors, and the combination of the two drugs has a synergistic effect on the treatment of breast cancer [118]. In addition, the PI3K/mTOR dual inhibitor gedatolisib combined with immune checkpoint inhibitors (ICI) more significantly inhibited the growth of breast cancer cells and continuously induced the responses of dendritic cells, CD8 + T cells, and NK cells, indicating that the inhibition of PI3K–AKT pathway may enhance breast cancer response to ICIs [119].

## 4. Natural Products and Synthetic Analogs for PI3K–AKT-Targeting Breast Cancer Treatments

Natural products from plants, mushrooms, and seaweeds are a source of drugs. Moreover, chemical drugs based on the molecular structure and functions of natural products have been developed. About two-thirds of anticancer drugs are derived from natural products and their derivatives [120]. Below, we discuss the research and development of natural compounds based in part on PI3K–AKT signaling targeting breast cancer, with emphasis on those in preclinical development (Table 2). The structures of these natural products and their mechanisms of action are shown in Figure 2 and Figure 3 respectively.

### 4.1. Flavonoids

#### 4.1.1. Curcumin

Curcumin, the most representative chemical polyphenol extracted from the rhizomes of turmeric, is used to treat various human disorders, including inflammation, metabolic syndromes, neurodegenerative diseases, and cancer. Studies have shown that curcumin inhibits breast cancer cell proliferation by targeting HER2-TK and NF-κB [136,137]. However, PI3K–AKT signaling has increasingly been shown to play an important role in the treatment of breast cancer with curcumin. In MDA-MB-231 breast cancer cells, curcumin-induced AMPK activation is involved in the activation of autophagy and the suppression of AKT levels, thereby inhibiting their proliferation and migration, impairing cellular uninterruptible power supply (UPS) function, accelerating Akt ubiquitination, and reducing Akt aggregation [127]. Additionally, curcumin has also been shown to induce breast cancer cell arrest in the S+G2/M phase and apoptosis, inhibit basic phosphorylation of AKT, and inhibit epidermal growth factor receptor (EGFR) and ERK1/2 phosphorylation induced by EGF pretreatment [128]. Curcumin also modulates the levels of p-Smad2 and β-catenin through transforming growth factor β (TGF-β) and PI3K–AKT signaling, thereby inhibiting doxorubicin-induced EMT [138]. Interestingly, the cytotoxic effects of curcumin on breast cancer cell lines have been shown to depend on their PI3K–AKT signaling status. Compared with MDA-MB-231 cells, a higher curcumin dose and treatment duration is needed against MCF-7 cells to maximize AKT phosphorylation and induce cytotoxicity [139]. Furthermore, Lv et al. found that curcumin can resensitize multidrug-resistant breast cancer to cisplatin via the inhibition of colon cancer associated transcript -1 (CCAT1) and PI3K–AKT signaling [140]. Notably, (2E,6E)-2,6-bis(4-hydroxy-3-methoxybenzylidene) cyclohexanone (BHMC), a curcumin derivative, also exhibits good anti-breast cancer effects in vivo.

#### 4.1.2. Quercetin

Quercetin is widely distributed in nature and is common in fruits and vegetables. Studies have shown that quercetin has a variety of pharmacological effects against human diseases, including antioxidant, anti-inflammatory, and antiproliferative effects [141,142]. Quercetin also inhibits breast cell proliferation and promotes apoptosis. In MCF-7 cells, quercetin has been shown to increase the ratio of bcl2-associated X protein/B-cell lymphoma 2 (Bax/Bcl-2) by reducing the expression of PI3K, AKT, ERα, and cyclin D1. It also significantly inhibits viability, clonal formation, and mammosphere generation in CD44+/CD24− breast cancer stem cells, and significantly suppresses in vivo tumor growth and metastasis of CD44+/CD24 cells [124]. Quercetin-3-methyl ether is reported to upregulate E-cadherin and to downregulate vimentin and matrix metallopeptidase 2 (MMP-2), thereby inhibiting the epithelial–mesenchymal transition (EMT) in breast cancer cells, while downregulating the expression of Notch1, PI3K, AKT, and enhancer of zeste homolog 2 (EZH2), which are important for the activation of breast cancer stem cells [143]. Compared with breast cancer cells treated with a combination of resveratrol, quercetin, and catechin, quercetin alone exhibits more obvious inhibition of proliferation and migration and arrests MDA-MB-231 and MDA-MB-435 cells in the G2/M phase of the cell cycle. Additionally, quercetin also markedly inhibits AKT activity, while enhancing the activity of AMP-activated protein kinase (AMPK), a negative regulator of mTOR. These effects suppress the activity of the downstream mTOR effector proteins, ribosomal protein S6 kinase beta-1 (p70S6K), and eukaryotic translation initiation factor 4E-binding protein 1 (4EBP-1) [144]. Interestingly, quercetin inhibits breast cancer development by suppressing cell migration and sugar degradation through the induction of AKT–mTOR signaling-mediated autophagy [145]. Esfandiar et al. found that the combination of quercetin and docetaxel upregulated p53 and Bax, downregulated Bcl-2, AKT, extracellular signal-regulated kinase 1/2 (ERK1/2), and signal transducer and activator of transcription 3 (STAT3), and enhanced the effect of docetaxel on the proapoptotic effect of MDA-MB-231 cells [146].

#### 4.1.3. Formononetin

Formononetin, a flavonoid compound isolated from Astragalus, has been investigated in recent years for its role in tumors and neurological diseases. Formononetin is a phytoestrogen and of the main components of clover [147]. Formononetin was reported to inhibit the activity of IGF1–PI3K–AKT pathways in a dose-dependent manner. Previously, it was reported that formononetin decreased the expression of cyclin D1, one of the downstream target proteins of AKT, which enhanced the G0/G1 phase in MCF-7 cells thereby decreasing proliferation. Moreover, formononetin treatment also inhibited tumor growth in xenografted human breast cancer cells in vivo [126]. It also significantly suppressed the proliferation of ER-expressing MCF-7 and T47D cells and promoted apoptosis of breast cancer cells by increasing (Ras), rapidly accelerated fibrosarcoma (Raf), and p-p38 expression, as well as the Bax/Bcl-2 ratio [148]. In addition to these effects, formononetin was demonstrated to downregulate the expression of MMP-2 and MMP-9 as well as increase the expression of tissue inhibitors of metalloproteinase-1 (TIMP-1) and TIMP-2. It inhibited the migration and invasion of MDA-MB-231 and 4T1 cells through the inhibition of the PI3K–AKT signaling pathway [149]. In a previous preclinical study, formononetin significantly abolished the Fibroblast growth factor 2 (FGF2)-stimulated human umbilical vein endothelial cell (HUVEC) proliferation. Formononetin inhibits tumor angiogenesis and growth by decreasing the activity of FGF2R, the phosphorylation of PI3K and AKT, and the activity of transcription factor STAT3 [150]. It is evident from the above study that the therapeutic effect of formononetin on breast cancer has clinical potential.

#### 4.1.4. Saponins

Ginsenosides are the main active substances isolated from ginseng. This plant is distributed in Northeast China, Korea, and Japan. American ginseng has also been found to contain ginsenosides. Recent studies have found that ginsenosides possess anti-proliferative, anti-metastatic, pro-apoptotic, anti-angiogenic, anti-multidrug resistance, and autophagy-regulating effects. Moreover, ginsenoside (Rg3) depolarizes mitochondrial membrane potential, releases cytochrome c, increases the expression of cleaved caspase-3 and cleaved PARP, the Bax/Bcl-2 ratio, activates the mitochondrial death pathway, and promotes apoptosis in breast cancer cells [151]. A study by Bo-Min et al. reported that ginsenoside (Rg3) inhibited the DNA-binding ability and transcriptional activity of NF-κB by reducing the expression of mutant p53 (R280K) in MDA-MB-231 cells. This was ascribed to the increased production of reactive oxygen species (ROS), the inhibition of ERK, and Akt phosphorylation by ginsenosides [134]. In addition to this, another ginsenoside (ginsenoside Rd) was found to play a similar role. Studies on MDA-MB-231 cells and HUVECs cells showed that ginsenoside (Rd) abolished vascular endothelial growth factor (VEGF)-induced VEGFR2 activation in HUVECs by decreasing intracellular Akt/mTOR/p70S6K and hypoxia-inducible factor 1-alpha (HIF-1α) activation. Moreover, ginsenoside (Rd) increased the expression of cleaved casepase-3 and other apoptosis proteins to inhibit angiogenesis and tumor growth in vivo and in vitro [133]. It has been reported that ginsenoside (Rk1) can also trigger S-phase cell arrest and induce apoptosis of MCF-7 cells [152]. Interestingly, Liu et al. found that ginsenoside (Rg5) caused less damage to the body compared with the first-line tumor treatment drug (docetaxel). The ginsenoside (Rg5) suppressed the phosphorylation levels of PI3K and AKT, and induced apoptosis and autophagy, thereby inhibiting breast cancer in vitro and in vivo models [153].

### 4.2. Non-Flavonoid Polyphenols

#### Resveratrol

Resveratrol is a non-flavonoid polyphenol, mainly found in grapes, peanuts, soybeans, and other plants, and also in Staphylococcus, Penicillium, Mucor, and other fungi. The estrogen receptor (ER) plays an important role in the development of breast cancer. Therefore, breast cancer is mainly treated by inhibiting the production of estrogen or blocking the binding of estrogen to its receptor. For this reason, resveratrol is used as a phytoestrogen to treat breast cancer [154]. Studies have shown that resveratrol inhibits the ERα but not ERβ [155]. In a study published in 2011, the combination of resveratrol and rapamycin increased the sensitivity to rapamycin and inhibited the growth of breast cancer cells by blocking the PI3K–AKT signaling pathway instead of the ERK- Mitogen-activated protein kinases (MAPK) signaling pathway. The downstream target of mTOR p70S6K was also inhibited leading to S-phase cell cycle arrest in breast cancer cells [156]. Interestingly, it has been found that resveratrol downregulated the expression of fatty acid synthase (FASN) and HER2 in a dose-dependent manner, and upregulated the expression of polyoma enhancer activator 3 (PEA3) (targeting the HER2 promoter to downregulate its transcriptional activity), thereby suppressing the proliferation of HER2-overexpressing breast cancer cells [157]. Previously, Kumar-Sinha et al. found that HER2 triggered FASN expression by activating the FASN promoter through the PI3K pathway [158]. Therefore, it is speculated that resveratrol decreases the proliferation of breast cancer cells by inhibiting the PI3K–AKT signaling pathway. Co-treatment of breast cancer cells with TGF-β1 and resveratrol upregulated E-cadherin, downregulated fibronectin, vimentin, Snail1, Slug, and alpha-smooth muscle actin (α-SMA), reversed TGF-β1-induced EMT through inhibition of PI3K–AKT and Smad signaling pathways [125]. In addition, resveratrol did not cause any side effects in mice. Evidence from previous studies also demonstrated that 3,5,4′-trimethoxystilbene (natural methoxylated analog of resveratrol) can reverse EMT by downregulating PI3K–AKT and Wnt/β-catenin signaling pathway cascades to inhibit invasiveness of breast cancer cells [159]. Collectively, these data show that resveratrol is a potential drug for the clinical treatment of breast cancer.

### 4.3. Others

Anthricin is a natural product isolated from *Anthriscus sylvestris* (L.) Hoffm. (Apiaceae), the root of which is often used as an antipyretic, cough suppressant, and pain reliever [160]. According to Chang et al., anthricin induces protective autophagy in breast cancer cells but promotes cell death by inhibiting autophagy. These effects are mediated by reduced the phosphorylation of AKT, p70S6K, and mTOR [130]. Piperine is an alkaloid found in black pepper (*Piper nigrum* L.) and is commonly used in India and China as a treatment for conditions such as intestinal disorders and epilepsy [161]. Studies have shown that piperine downregulates the expression of HER2 in SKBR3 cells. It can also reduce sterol regulatory element-binding transcription factor 1 (SREBP-1) activity by inhibiting the ERK1/2 signaling pathway, thereby decreasing the expression of FAS. Besides, piperine downregulates the transcriptional activity of NF-κB and activating protein-1 (AP-1) by blocking the Akt and MAPK signaling pathways. This abolishes the EGF-induced MMP-9 expression, and inhibits the proliferation and migration ability of breast cancer cells. Furthermore, the co-treatment of breast cancer with piperine and paclitaxel increased the sensitivity of cells to paclitaxel toxicity [132].

## 5. Limitations of Conventional Breast Cancer Treatment and Potential Biomarkers

The treatment of breast cancer is multidisciplinary, and early diagnosis and treatment may be an effective way to reduce breast cancer mortality [162]. Breast cancer diagnosis includes mammograms, as well as breast tissue biopsies, and multiple mammograms can cause potential radiation damage to the body [163]. Magnetic resonance imaging (MRI) is an effective auxiliary method, and its high sensitivity is helpful for the early diagnosis of breast cancer. However, the toxicity of gadolinium (Gd) metal in the contrast agent to the human body needs to be vigilant [164]. In addition, serological detection of breast cancer tumor markers CA 15-3, carcinoembryonic antigen (CEA), and CA 27-29 is one of the methods for auxiliary diagnosis of metastatic breast cancer. Current conventional treatments for breast cancer mainly include surgery, radiation therapy (RT), chemotherapy (CT), endocrine (hormone) therapy (ET), and targeted therapy [165]. Of these, breast-conserving therapy (BCT) and mastectomy are well-established local therapies for invasive breast cancer [166]. The selection of the treatment method that suits them is something that patients need to carefully consider before treatment. Up to 90% of women with breast cancer have long-term sequelae from treatment, including physical, functional, and psychosocial changes that can greatly affect the quality of life for breast cancer patients [167]. Breast-conserving therapy involves lumpectomy and postoperative radiation therapy, while connective tissue/collagen vascular disease and pregnancy may be contraindications to radiation therapy [166]. In addition, radiation therapy can include some side effects, decreased sensory function in the breast tissue and underarms, and itching, redness, and swelling of the skin at the surgical site [3]. Combining systemic adjuvant therapy may provide better help to patients according to their age, physical condition, and contraindications. For example, endocrine therapy (ET) may be more suitable for ER+ breast cancer, while CT is more recommended for ER− breast cancer and TNBC [165]. In the new adjuvant therapy, chemotherapy is often combined with some targeted drugs to treat breast cancer. The dysregulation of mTOR in TNBC in the PI3K–AKT pathway may increase the sensitivity of mTOR inhibitors in antitumor therapy [168]. Studies have shown that after paclitaxel combined with mTOR inhibitor Afinitor (everolimus) to treat TNBC for 48 h, the expression of mTOR is downregulated, the incidence of adverse events in patients with combined medication is not significantly increased, and the overall safety is good [169]. The role of the androgen receptor AR in breast cancer is gradually being uncovered, and AR is expressed in 90% of primary breast cancers and 75% of metastatic breast cancers [170]. Chia et al. showed that inhibition of AR resulted in decreased levels of phosphorylated Elk1 and c-FOS, as well as decreased ERK target proteins, in xenograft tumor models and patient tumors [171]. In addition, AR exhibits an important role after DNA damage in breast cancer cells, and dsDNA repair is significantly delayed after AR inhibition [172]. A preclinical study showed that AR can promote the invasion and migration of TNBC cells by activating Src signaling [173]. AR may be a potential biomarker in breast cancer treatment. In 15–30% of breast cancers, amplification and overexpression of HER2 occurs [174]. A study showed that HER2 can drive tumorigenesis independently of HER3 by activating the PI3K–AKT pathway [175]. As early as 20 years ago, the use of trastuzumab chemotherapy in HER2+ breast cancer achieved good results [176]. In recent years, new anti-HER2 treatment options have continued to be developed. For example, the novel anti-HER2 drug deruxtecan showed durable antitumor activity in a population of patients with previously treated HER2-positive metastatic breast cancer [177]. In addition, adding tucatinib to trastuzumab-treated metastatic HER2+ breast cancer can significantly prolong the progression-free survival in the trastuzumab and capecitabine combination therapy group [178].

## 6. Conclusions and Future Perspectives

Evidence from prior studies has demonstrated that impaired PI3K–AKT signaling contributes to the occurrence of many diseases, such as cancer and neurological diseases. The excessive activation of the PI3K–AKT pathway can lead to cancer development, whereas the inhibition of the PI3K–AKT pathway may lead to neuronal or glial cell death. Therefore, researchers should explore how to regulate the PI3K–AKT signaling pathway and its role in the pathogenesis of diseases.

In addition, much research into PI3K–AKT-targeting drugs is based on in vitro experiments or mice; thus, the transformation from basic studies to a clinical application must be screened and achieved. Many drugs or drug combinations have been shown to induce adverse events or toxicity in clinical trials leading to the termination of such trials. Based on the above problems, natural products may be candidates for the treatment of breast cancer. Natural products are widely distributed in vegetables, fruits, and medicinal materials in nature. Human beings have used natural products to treat diseases since ancient times. Diet alone may not be able to achieve the effect of natural products in treating diseases. To improve the applicability of natural products, the first thing we consider is the availability of natural products in the body. Preclinical studies are mainly performed using in vitro and mouse models, which are significantly different from the human body. In addition to screening for appropriate natural products, the optimization of natural products is also an urgent problem to be solved. Researchers have proposed that the use of substances such as turmeric oil, which is more easily absorbed, or combining curcumin with carrier substances such as nanoparticles can improve the efficacy of drugs. For example, combining curcumin with nanoparticles significantly improves the bioavailability of curcumin by increasing intestinal permeability, and stability in the microenvironment, reducing degradation, and extending half-life in blood [179]. Although its applicability needs further proof. In addition, the bioavailability of natural products can be increased through the chemical modification of their structures. For instance, the synthesis of ester derivatives of quercetin to bypass phase II metabolism during absorption potentially increased the systemic levels of quercetin [180]. In a previous study, piperine treatment inhibited breast cancer proliferation and migration, whereas the co-treatment of piperine and paclitaxel increased the sensitivity of breast cancer cells to paclitaxel [132]. A similar effect was reported when quercetin was used [146]. Therefore, drug combinations such as the TH combination (paclitaxel and trastuzumab) are more effective for the treatment of HER2-positive breast cancer. The efficacy of a drug may vary depending on the genotype of patients, and the application of precision clinical therapy is a challenge.

In preclinical experiments, natural products were used to treat breast cancer by targeting the PI3K–AKT pathway, and it was found that ERK, autophagy, UPS, and other signaling pathways were sometimes affected, and these signaling networks crossed each other. The role of natural products is complex. If natural products are further applied to the treatment of breast cancer, more research on their role is needed. Although natural products are promising anti-breast cancer drugs, cancer treatment is a complex process and we still rely on existing anticancer therapies. As mentioned above, natural products play a certain role in enhancing the sensitivity of cisplatin and paclitaxel. However, whether they also play a similar role in most drugs requires further research. It may be a good idea for the treatment of drug-resistant breast cancer. Natural products may play a more important role in tumor therapy by reducing drug toxicity, increasing its therapeutic concentration in the body, and reducing chemotherapeutic drug resistance.

Drug development requires collaborative efforts from many research teams including clinicians, and patients. At present, many studies have shown that natural products can inhibit the growth of breast cancer cells by regulating the PI3K–AKT pathway in mouse models. Most of the natural products discussed in this review have been studied in preclinical research. Their mechanisms in breast cancer treatment need to be further explored.

## Figures and Tables

**Figure 1 biomolecules-13-00093-f001:**
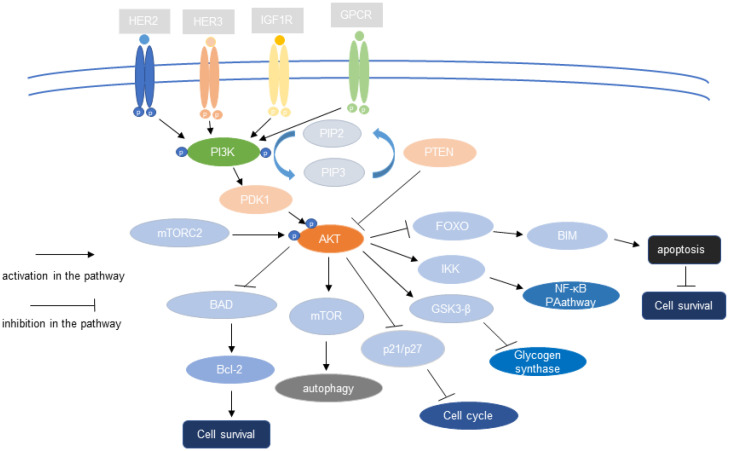
The PI3K–AKT pathway regulates tumor cell proliferation, cycle, autophagy, metabolism, and other cell biological behaviors through downstream effector molecules.

**Figure 2 biomolecules-13-00093-f002:**
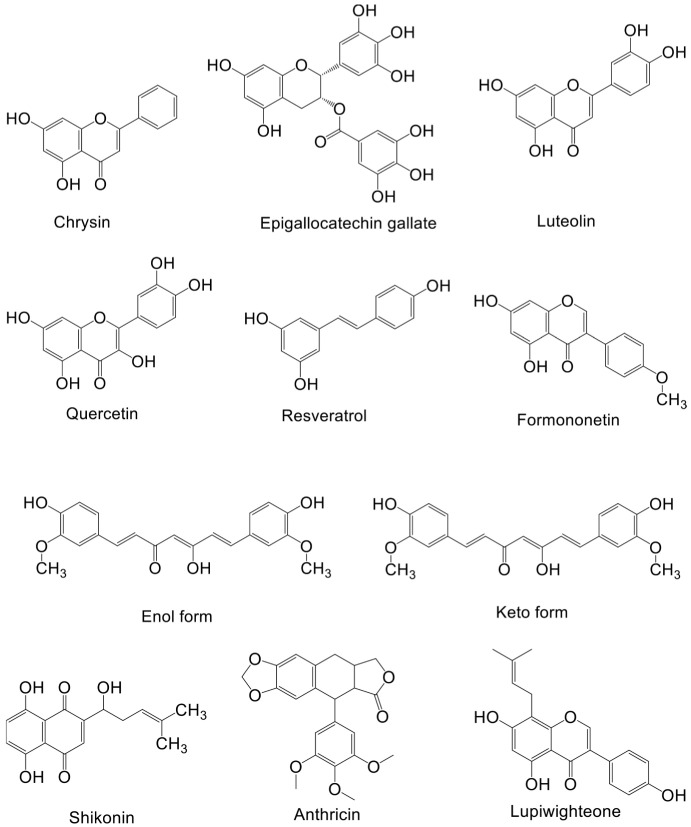
Structure of natural compounds against breast cancer.

**Figure 3 biomolecules-13-00093-f003:**
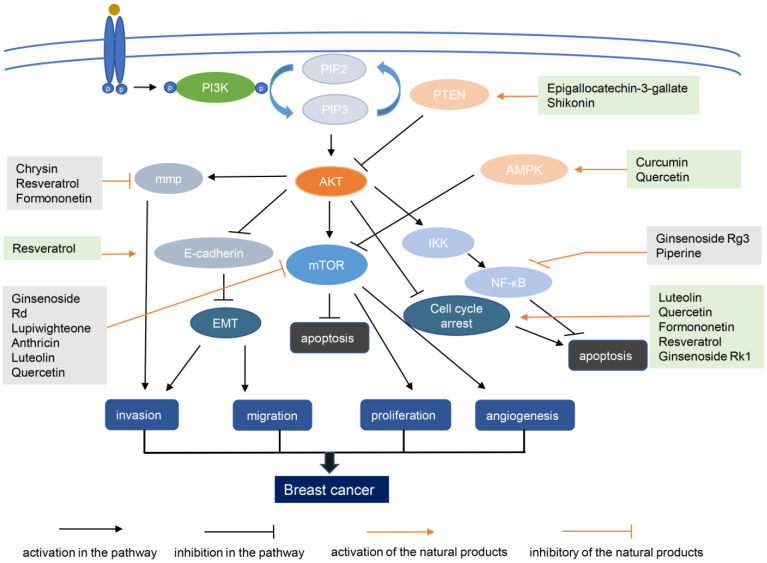
Mechanism of action diagram of natural products with potential for anti-breast cancer therapy. AMPK, amp-activated protein kinase; EMT, epithelial–mesenchymal transition; MMP, mitochondrial membrane potential; mTOR, mammalian target of rapamycin; NF-κB, nuclear factor kappa-light-chain-enhancer of activated B cells; PI3K, phosphoinositide 3-kinase; PIP2, phosphatidylinositol (4,5)-disphosphate; PIP3, phosphatidylinositol (3,4,5)-trisphosphate; PTEN, phosphatase and tensin homolog.

**Table 1 biomolecules-13-00093-t001:** Summary of PI3K–AKT-targeting drugs.

Drugs	Mechanism	Combination with	Phase	Refs.
BKM120	PI3K inhibitor	- Lapatinib Trastuzumab/Paclitaxel Fulvestrant Tamoxifen LDE225	Phase II Phase I/II Phase II Phase I Phase II Phase I	[58] [59] [60] [61] [62] [63]
Tenalisib (RP6530) Taselisib (GDC-0032)	PI3K δ/γ inhibitor PI3K inhibitor	- Enzalutamide	Phase II Phase I/II	[64] [65]
BYL-719 (alpelisib)	PI3K α inhibitor	- Letrozole Nab-paclitaxel	Phase II Phase I Phase I/II	[66] [67] [68]
Pictilisib (GDC-0941)	PI3K inhibitor	Cisplatin Paclitaxel (with and without Bevacizumab or Trastuzumab) and Letrozole	Phase I/II Phase I	[69] [70]
GDC-0084	PI3K inhibitor	Trastuzumab	Phase II	[71]
PF-05212384 (gedatolisib)	PI3K/mTOR inhibitor	Docetaxel/Cisplatin/Dacomitinib Paclitaxel and carboplatin	Phase I Phase I	[72] [73]
AZD8186	PI3K β Inhibitor	Docetaxel	Phase I	[74]
Serabelisib	PI3K α inhibitor	Canagliflozin	Phase I/II	[75]
Ipatasertib	AKT inhibitor	Trastuzumab and pertuzumab	Phase I	[76]
MK2206	AKT inhibitor	Lapatinib ditosylate Paclitaxel - Anastrozole (with or without goserelin acetate)	Phase I Phase I Phase II Phase II	[77] [78] [79] [80]

**Table 2 biomolecules-13-00093-t002:** Summary of some natural products for the treatment of breast cancer by inhibiting the PI3K/AKT pathway.

Drugs	Sources	In Vitro	In Vivo	Dose	Treatment Time	Mechanism	Refs.
Chrysin	Passiflora caerulea	MDA-MB-231 cell and BT-549 cell	-	5, 10, 20 μM	48 h	Inhibits p-AKT, vimentin and snail expression, inhibits metastasis	[121]
Epigallocatechin-3-gallate	Green tea	T47D cell and HFF cell	-	20, 40, 80 μM	72 h	Inhibits AKT and hTERT expression, induced apoptosis	[122]
Luteolin	Reseda luteola	MCF7-TamR cells	-	20, 30 μM	72 h	Inhibition of PI3K/AKT/mTOR, RAS expression, reverses tamoxifen-resistance in ER+ breast cancer cells	[123]
Quercetin	Capparis spinosa	MCF-7 cells and CD44+/CD24−CSCs	-	50 μM	24 h or 48 h	Inhibits PI3K/AKT/mTOR, promotes apoptosis, attenuates breast cancer stem cell cloning and mammary gland production	[124]
Resveratrol	Blueberries	MDA-MB-231, MDA-MB-453, MDA-MB-436, BT549 cells,	nude mice	25, 50 μM cell and 40 mg/kg mice	48 h or 72 h, 8 weeks	Inhibits PI3K/AKT/mTOR, decreases vimentin, slug, and MMP2, decreases cell viability and migration	[125]
Formononetin	Trifolium pratense	MCF-7 cells	nude mice	40, 80 μM cell and 15, 30, 60 mg/kg/day	48 h, 20 days	Inhibits cell proliferation, induces cell arrest in G0/G1 phase, reduce cyclin D1, p-IGF-1R, and p-AKT expression, inhibits local tumor growth in vivo	[126]
Curcumin	Curcuma longa	MDA-MB-231	-	25 μM	3 h, 6 h, or 24 h	Reduces the content of AKT protein, accelerates AKT ubiquitination, and affects AKT aggregation, impairs cellular UPS function, participates in the activation of autophagy, Inhibits the growth and migration of breast cancer cells	[127]
Curcumin	Curcuma longa	MDA-MB-468 cells and HBL100 cells	-	10 μM, 20 μM, 40 μM	24 h or 48 h	Reduces the phosphorylation of AKT and EGFR, inhibits the activities of ERK1, ERK2, and JNK, induces cell arrest in S and G2/M phases, triggers apoptosis of breast cancer cells	[128]
Shikonin	Lithospermum erythrorhizon	MDA-MB-231 and BT549 cells	-	1 μM, 5 μM	24 h	Decreases the expression of p-Akt, downregulates miR-17-5p, inhibits TNBC cell migration, invasion, and EMT, miR-17-5p binds to the 3’UTR of PTEN to downregulate its expression	[129]
Anthricin	A. sylvestris (L.) Hoffm	MCF-7 cell and MDA-MB-231 cell	-	25 μM, 50 μM	12 h or 24 h	Decreases phosphorylation of Akt, p70S6K, and mTOR, induces cell arrest in the G2/M phase, increased protective autophagy and apoptosis	[130]
Lupiwighteone	Glycyrrhiza glabra	MCF-7 cell and MDA-MB-231 cell	-	10 μM, 20 μM, 40 μM	48 h	Inhibits PI3K/Akt/mTOR signaling pathway, inhibits cell proliferation, induces caspase-dependent cell death	[131]
Piperine	Piper nigrum Linn	SKBR3 cell and MCF-7 cells	-	25 μM, 50 μM	24 h or 48 h	Reduces the expression of p-Akt, p-p38, and mmp-9 induced by EGF, enhances the effect of paclitaxel on breast cancer cytotoxicity, downregulates the expression of SREBP-1, and FAS and HER2, inhibits the migration of breast cancer cells	[132]
Ginsenoside Rd	Panax ginseng	HUVECs cell and MDA-MB-231 cell,	xenograft mouse	25 μM, 50 μM cell, 3 and 10 mg/kg/day mice	48 h,28 day	Inhibit Akt/mTOR/p70S6K and HIF-1α activation, prevent VEGF-induced HUVECs migration, invasion, and formation of capillary-like structures, reduces VEGF-induced VEGFR2 activation in HUVECs	[133]
Ginsenoside Rg3	Panax ginseng	MDA-MB-231 cell	-	30 μM	24 h	Inhibits the phosphorylation of ERK and Akt, reduces the transcriptional activity of NF-κB and the nuclear translocation of the p65 subunit, inhibits the degradation of IκBα and the catalytic activity of IKKβ, enhances the interaction between p53 and a negative regulator (Mdm2)	[134]
Notoginsenoside R1	Panax notoginseng	MCF-7 cell and MDA-MB-231 cell	-	75 μ, 150 uM	24 h or 48 h	Reduces p-PI3K and p-AKT levels, downregulates the expression of CCND2 and YBX3, and increase the cells arrested in the G1 phase, decreases YBX3 and expression of KRAS, inhibit the proliferation, migration, invasion, and angiogenesis ability of breast cancer cells	[135]

## Data Availability

Not applicable.

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
