# Peer review of "PI3K–AKT-Targeting Breast Cancer Treatments: Natural Products and Synthetic Compounds"

_biomolecules, 2023, doi:10.3390/biom13010093_

Round 1

Reviewer 1 Report

Comments

In this review paper, the authors have tried to discuss the role of the PI3K–AKT signaling pathway in the occurrence and development of breast cancer and highlight PI3K–AKT targeting natural products and drugs in clinical trials for the treatment of breast cancer.  The Review topic is interesting, and the whole manuscript is well structured. While this review work falls within the scope of biomolecules, it will have a significant impact on readers. It is good if the authors addressed the following topics before further consideration: -

1.     The authors must write a full name for each abbreviation in the manuscript at the beginning to minimize confusion for the reader. e.g., PI3K-AKT, PDGF, RTK, HER2, HER3, IGF1R, GSK-3, etc.…?

2.     Better to include, causes/risk factors, diagnosis, and currently available treatment methods, limitations of conventional treatment methods and exploring several targetable molecular biomarkers as an alternative or combinational breast cancer treatments

3.     It is good if the authors summarize using Figure/s for, PI3K–AKT signaling pathway overview and in particular the role of PI3K-AKT in breast cancer on pages 1, 2, or 3.

4.     It is good if the authors separately included the Upstream activation of the PI3K/Akt and downstream effector of the PI3K/AKT signaling pathway more in detail to treat breast cancer

5.     Although the authors have tried to summarize in the Table, it is highly recommended to incorporate more figures for each relevant section (especially for each application (in vitro or in vivo results) session).

6.     The authors discussed PI3K inhibitors and AKT inhibitors separately to treat breast cancer. However recently combination therapy is considered the best method to enhance the therapeutic efficacy of cancer, so it is good if authors add more information about the combination treatment of breast cancer by considering PI3K/AKT signaling inhibitors with other signal inhibitors or conventional therapies, such as chemotherapy, radiation therapy, and immunotherapy.

Reviewer 2 Report

In the present manuscript entitled “PI3K–AKT-targeting breast cancer treatments: potential drug 2 updates” the authors have nicely presented the involvement of the pathway in breast cancer including various synthetic or natural molecules targeting this pathway. This manuscript is short and sweet justifying the title and can be a good read for a novice researcher working on this target. There are a few suggestions that can be taken care of before the manuscript is accepted for publication. In line 142 the date on which the website was accessed for data collection can be included, this will provide a very clear idea of how old the data are. In line 143 the title should be extended to include the targeting drugs in different stages of clinical trials. As table 2 is focused on the natural molecules inhibiting the PI3K/AKT pathway, the word “activation” can be removed from the second last column of the table.

Reviewer 3 Report

The manuscript entitled “PI3K–AKT-targeting breast cancer treatments: potential drug updates” shows the perspectives of breast cancer target therapy using inhibitors of PI3K–AKT from natural sources. It is a very interesting and relevant paper, however, I would like to address to the authors some considerations.

1. English must be reviewed.

2. The meaning of all abbreviations (PIK3, PDGF, AKT, HER-2…) is missing. Even though these abbreviatures are common in breast cancer research fields, the meaning must be presented the first time when they appear in the text.

3. The focus on natural products could be explicit in the title

4. A Graphical Abstract could be provided (Figure 2 is a good start to a GA).

5. The introduction must be more robust.

6. Even in a narrative review, some criteria could be defined in the articles search.

7. In sections 3.1 and 3.2, the results could be more integrative or comparative between studies.

8. Table 2 must be revised. The comprehension of the data is not very easy. The division of the table into categories (cell type, the origin of the molecules, or in vitro/in vivo studies) could be a solution.

9. The results, mainly in section 4, must be discussed. There are few correlations between the showed results.
